# Orofacial Migraine or Neurovascular Orofacial Pain from Pathogenesis to Treatment

**DOI:** 10.3390/ijms24032456

**Published:** 2023-01-27

**Authors:** Yair Sharav, Yaron Haviv, Rafael Benoliel

**Affiliations:** 1Department of Oral Medicine, Sedation & Maxillofacial Imaging, School of Dental Medicine, Hebrew University-Hadassah, Jerusalem 91010, Israel; 2Unit for Oral Medicine, Department of Oral and Maxillofacial Surgery Division of ENT, Head & Neck and Oral and Maxillofacial Surgery, Tel Aviv Sourasky Medical Center-Ichilov, Tel Aviv 61060, Israel

**Keywords:** orofacial migraine, neurovascular orofacial pain, orofacial pain classification, orofacial pain pathophysiology

## Abstract

The purpose of the present study is to examine possible differences between orofacial migraine (OFM) and neurovascular orofacial pain (NVOP). Facial presentations of primary headache are comparable to primary headache disorders; but occurring in the V2 or V3 dermatomes of the trigeminal nerve. These were classified and recently published in the International Classification of Orofacial Pain, 1st edition (ICOP). A category in this classification is “orofacial pains resembling presentations of primary headaches,” which encompasses OFM and NVOP. The differences between NVOP and OFM are subtle, and their response to therapy may be similar. While classified under two separate entities, they contain many features in common, suggesting a possible overlap between the two. Consequently, their separation into two entities warrants further investigations. We describe OFM and NVOP, and their pathophysiology is discussed. The similarities and segregating clinical signs and symptoms are analyzed, and the possibility of unifying the two entities is debated.

## 1. Introduction

Facial presentations of primary headache disorders were recently reviewed [1] and largely relied on previously published case series and reports. More recently, facial pain resembling migraines or the Trigeminal Autonomic Cephalalgias (TACs) have been reported in a large patient cohort [2]. These facial representations are not easy to diagnose as they appear in the lower two-thirds of the face, frequently in the maxillary region, the upper and lower jaws, including the teeth. Phenotypically they often most closely resemble sinus pain or toothache.

The accumulating data have led to the establishment of an innovative group of orofacial pain (OFP) whose major feature is that they are comparable in various parameters to primary headache disorders occurring in the V2 or V3 dermatomes. These were classified and recently published in the International Classification of Orofacial Pain, 1st edition (ICOP) [3]. A category in this classification (Section 5) is “orofacial pains resembling presentations of primary headaches”. This category covers four main sub-sections: orofacial migraine, tension-type orofacial pain, trigeminal autonomic orofacial pain, and neurovascular orofacial pain. Other major categories in ICOP include classifications of dentoalveolar pain, pain from the TMJ, the masticatory muscles, idiopathic pain and pain resembling primary headache, the latter being the focus of this review. ICOP was researched and designed with the principles underlying the International Classification of Headache Disorders (ICHD-3). In many ways, therefore, it is not only a classification of OFPs but a true bridge between the intimately related topics of head and face pain. ICOP underscores that in clinical practice we often see three types of patients who seem to typify the intersection between headache and orofacial pain (OFP). Type 1: Headache patients who report additional facial pain during, and usually ipsilateral to, the headache attacks. Type 2: Headache patients whose headache attacks have stopped and been replaced by facial pain attacks of the same quality, length, and intensity, including occurrence of the associated symptoms of the former headache. Type 3: Headache naive patients who develop de novo OFP attacks that resemble one of the primary headache types in pain character, duration, and intensity, with or without the associated symptoms of these headache types.

It seems that ICOP, for the purpose of “pure” classification includes under orofacial migraine (OFM) or neurovascular orofacial pain (NVOP) only patients in the third category, who have de novo pain exclusively in the facial region *but with no head pain*. Yet, in clinical practice we often see patients with a history of migraine, who suddenly develop severe tooth ache that does not respond to conventional dental treatment but treated successfully with antimigraine medications. Or patients who sometimes has “conventional head located” migraine, and on other occasions a migrainous toothache. These are often spontaneous or triggered. We believe that eventually these patients should be included and studied in clinical studies of OFM or NVOP as has been reported [4,5]. As in all classifications, ICOP will develop and change as data are collected and published. ICOP very much reflects the first version of ICHD, as it was in the 1980s.

Our paper will primarily relate to orofacial migraine (OFM) and neurovascular orofacial pain (NVOP). Particularly, we will discuss possible similarities and differences between the two, and whether these are two separate entities or should be merged into one entity; bearing in mind that their response to therapy is very similar [1,6,7,8,9,10,11]. Yet, there are subtle differences between NVOP and OFM and their characteristics need further careful research. This will further define their independence or further elucidate their relationship on the one hand and NVOP and the trigeminal autonomic cephalalgias (TACs) on the other.

## 2. Pathophysiology

At the core of the response to the question as to whether OFM and NVOP are separate lie specific pathophysiologic features. It is thought that migraine headache is a manifestation of a brain state of altered excitability capable of activating the trigeminovascular system in genetically susceptible individuals [12]. Advances in in-vivo and in-vitro technologies indicate that cortical spreading depolarization (CSD) and activation of the trigeminovascular system and its constituent neuropeptides, as well as neuronal and glial ion channels and transporters, contribute to the putative cortical excitatory/inhibitory imbalance that renders those with migraine susceptible to an attack [13].

The mechanisms underlying facial pain presentations of headache disorders remain unknown. On the one hand there is really little reason for scientifically separating the head and face. Although complex, they are connected extensively. Attempts at redefining their anatomical separation with sharp anatomical lines [14] are useful in learning anatomy but clinically will continue to fail us. This is supported by the all-encompassing innervation of the intra- and extra-cranial innervation by the trigeminal nerve.

There is direct anatomical communication between the intra- and extracranial innervations of the trigeminal nerve: In both rat and human dura mater, some intracranial fibers leave the skull through emissary canals and fissures to innervate the periosteum and extracranial tissue such as the pericranial muscles [15]. Therefore, the anatomical connection between the intracranial and extracranial fibers provides a route of how trigeminovascular activation of the dura extends to their extracranial counterparts, the dermatomes in the face [16]. The intracranial structures for pain perception, i.e., the dura mater, are primarily innervated with the V1 branch. However, the dura mater in the posterior cranium is innervated by V2, V3 and cervical branches [17]. Intracranial activation of V2/V3 fibers is therefore more likely to evoke posterior and lower face pain, whereas intracranial activation of V1 would evoke frontal or facial pain. Extracranial activation of the trigeminal nerves may also lead to the intracranial activation of their counterparts. Neurogenic inflammation via intranasal administration of capsaicin and formalin increased plasma protein extravasation not only in the nasal mucosa, but also the dura mater [18]. Based on the anatomical and functional connections between different branches of the trigeminal system, it is surprising that facial presentation of headache disorders remains so rarely reported and we believe that it is more likely it is unrecognized. 

When peripheral anatomy remains insufficient to explain a low prevalence of facial pain presentation, such somatotopic segregation may be rather central. The central somatotopy of trigeminal nucleus caudalis (sTN) is onion-ring shaped with the center being the perioral region. However, fibers from the V1 branch project more to the caudal part of the trigeminal nucleus caudalis (sTN), whereas those from the V2 and V3 branches more to the rostral part of the sTN [19]. This distribution also provides the anatomical basis of why cervically targeted therapies, e.g., greater occipital nerve (GON) block, may be effective in aborting headache disorders, since the V1 dermatome projected to the most caudal part of the sTN and is located directly adjacent to the secondary sensory neuron of the C2/C3 branches in the spinal cord [20,21]. It has recently been demonstrated that the stimulation of the V1 dermatome via capsaicin was able to modulate the pain threshold in the V2, V3, and GON dermatome; similarly, stimulation at the GON was able to change the pain threshold on all three branches of the trigeminal nerve, but with a stronger effect on V1, compared to V2/V3 [22]. This study provided evidence that the functional interaction between different branches of the trigeminal nerve takes place at the pontomedullary level and would suggest that GON blocks may be less effective for pain modulation in the lower facial region. It has been demonstrated that the functional connection between the limbic system and the ophthalmic branch exist in migraine and explains the attack-like behavior [23,24] this functional connection establishes a neuroanatomical basis for attack-like pains in the head [25]. Similar evidence would be useful in explaining pain in the V2 and V3 dermatomes. Hypothetically the facial presentations could be a simple “spread” of the pontomedullary activation in type 1 and type 2 facial presentations of headache, whereas the isolated facial attacks resembling headaches (type 3) are due to a (extremely rare) direct functional connection between the limbic system and the maxillary or mandibular brainstem nuclei. Further studies into this subject are clearly needed.

Functional imaging studies for headache and facial pain disorders suggested possible different mechanisms behind headache and facial pain. Brain activation in the sTN via trigeminal nociception was decreased in migraine [26] but increased in primary facial pain disorders (i.e., persistent idiopathic facial pain) [27] suggesting a role of hyperresponsive secondary sensory neurons in facial pain. Patients with an orofacial presentation of primary headache disorders have yet to be investigated using neuroimaging methods.

## 3. Orofacial Migraine

According to the International Classification of Orofacial Pain (ICOP) [3], orofacial migraine is subdivided into episodic and chronic types. Both occur exclusively in the orofacial region, without head pain, with the characteristics and associated features of migraine as described in ICHD-3 [28]. The episodic type (Table 1) is characterized by recurrent attacks, lasting 4–72 h. Typical characteristics of the pain are unilateral location, pulsating quality, moderate or severe intensity, aggravation by routine physical activity and association with nausea and/or vomiting photophobia and phonophobia. The chronic type (Table 2) has the characteristics of the episodic facial and/or oral pain, occurring on 15 or more days per month for more than 3 months, and which has the features of migraine on at least 8 days per month.

There have a been several reports on migraine-like pain in the lower two thirds of the face. With no clear diagnostic criteria, at the time, different terms were assigned, such as: orofacial pain with vascular-type features, orofacial migraine, lower half migraine, migraine with isolated facial pain, or migraine presenting as isolated facial pain [5,8,9,29,30,31,32]. Although some authors chose to refer to their study as “migraine with/or presented as isolated facial pain”, all patients in the Obermann et al. [9] study presented toothache, mostly reminiscent of pulpitis (acute tooth pulp inflammation), and 65% of Lambru et al. patients [5] underwent tooth treatment; implying suspected intraoral pain location. We therefore discuss their findings also under NVOP. 

Facial presentations of primary headache disorders are considered rare particularly isolated orofacial migraine. A population-based study investigated a sample of 517 people with migraine and found that in 46 cases (8.9%) migraine pain was focused in the head but extended to the lower half of the face [33]. Only one patient was identified by Yoon et al. with isolated facial migraine. However, Yoon et al. [33] admitted that this low rate could be due to a biased sampling; and those “*having isolated facial pain without any other migraine symptoms could have been neglected*”. The phenomenon, still poorly recognized, was not new. Lance mentioned lower half headache under vascular headache of migraine type to separate this group from atypical facial pain [34]. Additionally, 6% of 973 patients in a head and neck practice setting described migraine-associated pain isolated to the second trigeminal division [8], and of 100 patients with ‘sinus headache’ 85% had migraine or probable migraine and 1.6% reported pain confined to the second trigeminal division [35]. Recently, Lambru et al. [5] found that out of 1176 patients with migraine, 58 were defined as isolated facial migraine. At this early stage, establishing a prevalence is unreliable—5% of migraine patients [5] would seem a reasonable interim prevalence to work with.

It is surprising that this presentation has caused diagnostic difficulties. Careful examination of the ICHD-3 criteria for migraine reveals that the location of pain is unspecified except for describing it as “unilateral” [28]. In the footnotes for migraine the following appears; “a subset of otherwise typical patients has facial location of pain, which is called ‘facial migraine’ in the literature; there is no evidence that these patients form a separate subgroup of migraine patients.”

Ours and the experience of others further suggests, that in addition to an orofacial component in migraine attacks, orofacial pain can be totally isolated from head pain [5,8,29,30,31,32]. Moreover, often these isolated facial pains present with a clinical phenotype that, other than the location, may be diagnosed as a migraine or TAC variant. Research has shown that, in addition to the facial location, patients with isolated orofacial migraine report significantly more trigemino-autonomic signs; conjunctival injection, tearing, rhinorrhea, miosis, ptosis, eyelid oedema, nasal congestion and facial flushing than in other migraine patients (47.8% vs. 7.9%; *p* < 0.001) [5,33]. The unusual location of autonomic signs, both in migraines and TACs, often leads to erroneous diagnoses relevant to our discussion, such as oral pathology or sinusitis [7,8,10,29,30,32]. 

As clinicians, the essence of diagnosis is therapy. Very important to appreciate that these facial presentations of headache disorders, with mixed migraine and trigeminal autonomic characteristics, that are often misdiagnosed are repeatedly mistreated as dental or rhino nasal problems [5,7,31].

Since the management of OFM is like that of NVOP, treatment options will be discussed together for both entities.

## 4. Neurovascular Orofacial Pain

In our 1997 study [32], we were able to collect patients with orofacial pain and apply the then-current ICHD criteria, allowing for an atypical location in the migraines and TAC like pains. We were able to identify that many patients, although pain was atypically located, displayed the features of cluster headache, paroxysmal hemicrania and migraine [32]. However, a group with primary facial pain exhibiting neurovascular characteristics but not fitting any of the existing diagnoses was identified [32]. Although theoretically an atypical form of orofacial migraine/TAC the features were extremely mixed and represented an entity separate from the migraines or TACs (Benoliel et al., 2008; Benoliel et al., 1997) [29,32]. Moreover, pain was typically tooth-located and aggravated by cold food or beverages, very similar to teeth affected by a carious lesion; except that these teeth were intact. We later have termed this entity “neurovascular orofacial pain” (NVOP) [29].

**Clinical Features**. Table 3 summarize the definitions of NVOP by the ICOP^3^, first described by us in 1997 [32], followed by additional reports [1,6,7,29,36], up to our most recent study [4].

**Location.** The vast majority of patients report unilateral pain (76%), ignoring Benoliel et al. [32] and Obermann et al. [9] data, who included unilateral cases only (Table 4). Pain occurs primarily intraorally, teeth are often affected, around the alveolar process (62%) and adjacent mucosal sites (32%) [31,32]. In 35% of cases pain referral was to perioral structures (lips, chin, etc.), to the periorbital region (usually infraorbital) in 35% and to the preauricular region in 30%. Pain location is typically different from that described for migraine. In many publications, the primary site affected is the malar or infraorbital region [2,5,32,35]. Site of pain is a strong driver of diagnosis. Patients initially choose whom to consult based on pain location and location is a major anamnestic factor in the diagnostic process.

**Quality and Temporal Pattern.** NVOP is characterized by strong pain (7–8 on VAS), pulsating and episodic. Pain may last from minutes to hours, and up to 3 days [5]. Many cases are characterized by a high frequency daily pattern of spontaneous pain or evoked by cold food ingestion. In both instances, orofacial migraine and NVOP are subdivided each into episodic and chronic types. Of which 60% of cases are episodic in nature (see Table 4).

**Accompanying Phenomena.** NVOP can be accompanied by various local autonomic signs (AS), and these were found in close to 80% of cases (Table 4). Specifically tearing (10–20%), conjunctival injection (14%), miosis (14%), ptosis (3%), nasal congestion (7–40%), a feeling of facial redness or swelling (3–7%), and a complaint of excessive sweating (7%) were reported [5,32]. Other phenomena such as photo- or phonophobia (14%) and nausea (24%) were observed [9,32]. Often patients report dental hypersensitivity to cold, leading to diagnostic confusion [6,7]. Pain may be aggravated by physical activity [9].

NVOP is of importance in the differential diagnosis of orofacial pain to avoid misdiagnoses as sinusitis and in particular dental pathology. In addition to the location outside the conventional boundaries of migraine and TACs, NVOP presents with a distinctive combination of clinical signs and symptoms, i.e., high sensitivity of teeth to cold application [4,32]. Thus, the rationale for introducing NVOP is based on specific features that segregate it from other primary neurovascular-type craniofacial pain [1].

**Epidemiology.** The onset of NVOP is around 40–50 years of age (mean 43.4 years), with a female/male ratio approaching 4:1 [4,5,9,29,32], (see Table 4). Time to diagnosis was around 34–101 months (range 1–528 months) attesting to the diagnostic difficulties presented by these patients [31,32]. In 30–65% of cases, the pain was diagnosed as secondary to dental pathology and patients underwent dental treatment with no success [5,6,31,32].

A population-based study demonstrated that facial pain was not unusual in migraine (8.9%), yet *isolated* facial migraine was exceptionally rare (0.2%) [33]. However, Yoon et al. [33] were aware of some limitations of their study. Their screening question studied only migraine sufferers with respect to additional or isolated facial pain. Therefore, those having isolated facial pain without any other migraine symptoms could be neglected. Indeed, it has been our experience that many of our NVOP patients do not have a history of migraine. Recently, Lambru et al. [5] found that out of 1176 patients with migraine, 58 were defined as isolated facial migraine. Their pain location was restricted to intra- or/and extraoral areas, and 65% of these patients underwent endodontic treatments or multiple dental extractions featuring this group very much akin to NVOP patients. Thus, isolated facial migraine (and/or NVOP) accounted for about 5% of their migraine patients. With the prevalence of migraine of 15.3% (males 9.7%, females 20.7%) in the US adult population [37], and the preponderance of females in the NVOP group (about 80%), we estimate that the prevalence of NVOP approximate about 1–2% of the population. As stated previously, this seem a well-founded provisional prevalence to use until further data gather.

## 5. Management

Low dose amitriptyline, propranolol and anti-convulsant therapy have been a successful prophylactic strategy in NVOP patients [5,6,7,31,32]. Topiramate, an anti-convulsant, is a very effective prophylactic agent for chronic migraine [38]. Based on our experience topiramate was very effective in the management of NVOP; particularly for the chronic type. Recently a new class of anti-migraine therapy, Calcitonin Gene Related Peptide (CGRP) receptor antagonists -gepants: (ubrogepant, rimegepant, atogepant) and anti-CGRP monoclonal antibodies (erenumab, fremanezumab, galcanezumab and eptinezumab) were developed for prophylactic as well as abortive treatment of migraine [39]. None was systematically used for treatment of OFM or NVOP. Some studies indicate the feasibility of the use of these medications in facial pain [40,41]. We therefore recommend the trial of these substances particularly in resistant cases of OFM or NVOP. Triptans as abortive agents were reported in one study and was effective in all patients [9]. While abortive or prophylactic treatments could be considered, it has been our experience that prophylactic treatment is generally indicated in most NVOP patients. We recommend the prophylactic mode of therapy for almost daily pain, characterized by a high frequency pattern of spontaneous nature or evoked by cold food ingestion. Yet, a response to triptan [9] abortive treatment, which affects only migrainous pain, helps to distinguish between pain of dental or NVOP nature, especially in patients with ambivalent findings of a complex nature. It is usually recommended that patients with suspected NVOP should be referred to a practitioner specializing in Orofacial Pain.

Orofacial migraine located out of the oral cavity, and not aggravated by cold food ingestion, should be treated according to frequency and/or chronicity. Abortive (e.g., triptans) [11] versus prophylactic (e.g., amitriptyline, topiramate) therapy should be considered accordingly.

## 6. Discussion

The International Classification of Orofacial Pain (ICOP) [3] divided orofacial migraine (OFM) and neurovascular orofacial pain (NVOP) into two discrete categories. We summarize the characteristics of OFM and NVOP in Table 5; examining the features that are common to both entities and those that are unique. We examined whether these are two separate entities should be merged into one entity; bearing in mind that their response to therapy is very similar [1,6,7,8,9,10,11]. Clinically, there are subtle differences between NVOP and OFM and their characteristics need further careful research, as summarized in Table 5.

ICOP includes under OFM or NVOP only patients who have de novo pain exclusively in the facial region *but with no head pain*. In clinical practice, we often see patients who sometimes have “conventional head located” migraine, and on other occasions facial migraine or NVOP. We therefore included in our review studies that include patients with OFM or NVOP regardless whether they had de novo symptoms or those associated with “conventional” migraine.


**Common features for OFM and NVOP**


**Location.** By definition, OFP is facial and/or oral, and NVOP in intraoral; both defined as *unilateral*. However, NVOP may radiate to adjacent sites expanding beyond the intraoral definition. Side shift may occur, although pain is mostly unilateral, and bilateral cases are reported in up to a third of cases of NVOP or OFM.

It is therefore obvious that the location of OFM and NVOP are very similar.

**Time course.** OFM lasts 4–72 h (episodic type) and NVOP lasts 1–4 h (short lasting), but it may also be >4 h (up to, not specified), and then it is defined as long-lasting. It seems that the time course does not delineate OFM from NVOP because long-lasting NVOP can be as long as OFM.

**Intensity.** Both, OFM and NVOP are of moderate to severe intensity.

**Associated signs and symptoms.** Pulsating quality, nausea and/or vomiting and photophobia and phonophobia are common to both.


**Unique features for OFM and NVOP**


**OFM.** Aggravation by, or causing avoidance of, routine physical activity (e.g., walking or climbing stairs). But one should notice that this is not mandatory as it can be an option chosen out of several signs and symptoms.

**NVOP.** Toothache-like and autonomic signs. However, one should notice that these are not mandatory as they can be an option chosen out of several signs and symptoms.

## 7. Conclusions

The differences between NVOP and OFM are subtle, and their response to therapy is similar. It should be noted, however, that NVOP necessitate mostly prophylactic treatment, while OFM can in many instances be treated abortively. Consequently, their separation into two entities warrant further investigations. Presently, for research purposes, the separation into two entities may be justified. Which permits a more precise diagnostic definition and enable better communication between investigators. With time, as more data accumulate the justification for separating these two entities or merging under one diagnosis may become clearer.

## Figures and Tables

**Table 1 ijms-24-02456-t001:** Diagnostic Criteria for Episodic Orofacial Migraine. (Adapted from [3]).

Diagnostic Criteria		Notes and Comments
A	At least five attacks fulfilling criteria B–D	
B	Facial and/or oral pain, without head pain, lasting 4–72 h (untreated or unsuccessfully treated)	Episodic orofacial migraine, as defined (with no head pain), seems to be very rare. Bilateral orofacial migraine has not so far been described.
C	Pain has at least two of the following four characteristics: 1. unilateral location 2. pulsating quality 3. moderate or severe intensity 4. aggravation by, or causing avoidance of, routine physical activity (e.g., walking or climbing stairs)	Orofacial migraine with aura has not, to our knowledge, been described, and is excluded from ICOP until better evidence of it accumulates.
D	Pain is accompanied by one or both of the following: 1. nausea and/or vomiting 2. photophobia and phonophobia	A group of patients with attacks of intraoral pain of varying duration, with atypical migraine-like features, have been described. These may be unrelated to migraine, and are described under Neurovascular orofacial pain.
E	Not better accounted for by another ICOP or ICHD-3 diagnosis	

**Table 2 ijms-24-02456-t002:** Diagnostic Criteria for Chronic Orofacial Migraine. (Adapted from [3]).

Diagnostic Criteria		Notes and Comments
A	At least five attacks fulfilling criteria B–D Facial and/or oral pain, without head pain, on >_15 days/month for >3 months and fulfilling criteria B and C below	Characterization of frequently recurring OFP generally requires a pain diary to record information on pain and associated symptoms day-by-day for at least 1 month.
B	Occurring in a patient who has had at least five attacks fulfilling criteria B–D for Episodic orofacial migraine	
C	On >_8 days/month for >3 months, fulfilling either of the following: 1. criteria C and D for 5.1.1 Episodic orofacial migraine 2. believed by the patient to be orofacial migraine at onset and relieved by a triptan or ergot derivative	
D	Not better accounted for by another ICOP or ICHD-3 diagnosis	

**Table 3 ijms-24-02456-t003:** Diagnostic Criteria for Neurovascular orofacial pain (NVOP). * (Adapted from [3]).

Diagnostic Criteria		Notes and Comments
A	At least five attacks of unilateral intraoral pain of variable duration, without head pain, fulfilling criteria B–D	Although essentially an intraoral pain, there may be referral and/or radiation to adjacent sites, particularly when pain is severe.
B	Pain has both of the following characteristics: 1. moderate or severe intensity 2. either or both of the following qualities: (a) toothache-like (b) pulsating	Side shift may occur, although pain is mostly unilateral, bilateral cases are reported in up to a third of cases.
C	Pain is accompanied by at least one of the following: 1. ipsilateral lacrimation and/or conjunctival injection 2. ipsilateral rhinorrhea and/or nasal congestion 3. ipsilateral cheek swelling 4. photophobia and/or phonophobia 5. nausea and/or vomiting	There are reports of abnormal sensitivity to cold, both interictally and during attacks.
D	Pain is unexplained by any local cause, and clinical and radiographic examinations are normal	Frequently painful vital teeth will be hypersensitive to cold stimuli. Some of the teeth in the painful region may have undergone root canal therapy with no long-lasting pain relief
E	Not better accounted for by another ICOP or ICHD-3 diagnosis	

* Subdivided into short-lasting and long-lasting as follows: Short-lasting neurovascular orofacial pain: Attacks of intraoral pain fulfilling criteria for Neurovascular Orofacial Pain, and lasting 1–4 h (untreated, or unsuccessfully treated); Long-lasting neurovascular orofacial pain: Attacks of intraoral pain fulfilling criteria for Neurovascular Orofacial Pain, and lasting >4 h.

**Table 4 ijms-24-02456-t004:** Demographics and pain characteristics of NVOP and isolated facial migraine.

	Haviv et al. 2020 [4]	Lambru et al. 2020 [5]	Benoliel et al. 1997 [32]	Benoliel et al. 2008 [29]	Obermann et al. 2007 [9]	Gaul et al. 2007 [30]	Weighted Average
**Pain definition**	NVOP	Isolated facial migraine *	NVOP	NVOP	Isolated facial migraine ***	Orofacial migraine ***	
**Number of subjects**	80	58	29	23	7	2	
**Age**	39.8 +/− 12.6	49 +/− 9.9	42.6 (17–66)	39 +/− 13.7	55.4 +/− 3.2	46	43.4
**Females**	79.3%	79%	75%	70%	86%	100%	78%
**Pain location**	Oral 34.5% Perioral 65.5%	V^2^ 85% V^2^–V^3^ 10% V^3^ 5%	Intraoral ** (62% Dental)	Oral and perioral	V^2^–V^3^	Dental	
**Unilateral**	67.5%	79%	93% **	70%	100% **	100%	76%
**Bilateral**	32.5%	16%		30%			26.2%
**Pain intensity (VAS)**	8.1 +/− 1.5	7–10	Severe	8.3 +/− 1.4	8 (6–10)	8	Severe 7–8
**Episodic pain**	55%	66%	100% **	30%	100%	100%	59.2%
**Attack duration**	Hours-days	4 h–3 days	Mins–hours	11.1 +/−13.2 h	N.A.	Half to 1 day	
**Chronic pain**	48.8%	34%		70%			40.8%
**Autonomic/** **systemic sings**	78%	96%	55%	65%	86%	100%	79%
**History of migraine**	Excluded from study	58/1176 (5%) out of migraine subjects	Excluded from study	N.A.	N.A.	N.A.	
**Successful treatment modalities**	N.A.	Triptans and prophylact. anti-migraine medication	N.A.	N.A.	Triptans and prophylactic anti-migraine medication	Triptans	
**Attempted dental intervention**	36.3%	65%	38%	30%	None	None	

* Pain located at Intra- or/and extraoral areas; ** Inclusion criteria of study and therefore excluded from weighted average; *** Toothache initial complaint; N.A. = not available.

**Table 5 ijms-24-02456-t005:** Common or unique features for orofacial migraine (OFM) and neurovascular orofacial pain (NVOP).

	OFM	NVOP
**Location**	intraoral and unilateral *	facial and/or oral and unilateral *
**Time course**	NVOP lasts 1–4 h (short lasting), but it may also be >4 h (up to- not specified), and then it is defined as long-lasting. **	lasts 4–72 h (episodic type) Chronic OFM was defined as at least 15 days a month **
**Intensity**	moderate to severe intensity	moderate to severe intensity
**symptoms**	Pulsating quality, Toothache-like ^1^	Pulsating quality
**Associated signs**	nausea, photophobia/phonophobia. Autonomic signs ^1^; tearing, conjunctival injection	nausea and/or vomiting, photophobia and phonophobia, aggravation by physical activity

* NVOP may radiate to adjacent sites. Side shift may occur, although pain is mostly unilateral, bilateral cases are reported in up to a third of cases. It seems that locations are similar for OFM and NVOP. *Except that OFM must be unilateral.* ** It seems that the time course does not delineate OFM from NVOP because long-lasting NVOP can be as long as OFM. ^1^ Toothache-like and autonomic signs, are not mandatory as they can be an option chosen out of several signs and symptoms.

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
