# Peer review of "Orofacial Migraine or Neurovascular Orofacial Pain from Pathogenesis to Treatment"

_ijms, 2023, doi:10.3390/ijms24032456_

Round 1
Reviewer 1 Report
Dear authors, thank you for this very well written paper. It was a pleasure to read. This topic has caused confusion in the orofacial pain literature, and I found it enlightening. Some minor changes are suggested.
Abbreviations are used in the first mention of certain entities, such as TACs in the introduction. The conclusion rather than being placed at number six should follow the end of the paper. Tables 1 and 2 should be clearly titled so that the reader is aware it is from ICOP 1.
Could you please clarify under the treatment section if you mean treatment or management? I am also not clear on whether response to triptans is helpful in distinguishing between orofacial migraine and neurovascular orofacial pain. Line 276 is confusing. In this section, could you please also comment on whether specialist referral is recommended (or something that would help a general dentist reading this paper decide on what to do next if they suspect this for their patient?)
Table 5: OFM must be unilateral, could you please clarify if it is sidelocked?
There are some minor grammatical and punctuation errors.
Thank you again.
Reviewer 2 Report
This manuscript covers an interesting and important topic for the general practitioner and dentist. The authors have done a good job, but in some sections/points they need to carefully improve the manuscript.
- From the title the reader expects a detailed description of the two diagnoses, but in reality, the whole manuscript seems like a brief review; perhaps the title should be improved by better emphasizing the scope of their work.
- The purpose or objectives are not clearly described and a major English overhaul is needed.
- At the end of the introduction section, it is said “…we will discuss…”, but when reading this section (Discussion) does not exist.
- In general, the authors create confusion with the division of sections and subsections. They need to reevaluate their order; for example, section number 6 (Conclusions) is before the last two sections and includes a table, which makes no sense. Conclusions should be at the end of the manuscript and clearly mention future results and outcomes.
- It is important to keep the same acronym after the first time that the respective full name is mentioned, this is not seen kept, and creates confusion; for example, TACs is used as an acronym but is not its full name mentioned above.
- In several places there are unnecessary expressions or sentences in italics; in addition, references must be formatted according to journal guidelines.
- Table 4 is not clear how it was created; the references (authors) of the table must be reordered from the latest to the old publication and reformatted it (on the left of the table put the references and above the components).
Round 2
Reviewer 2 Report
The authors have improved their manuscript after the comments made!